# Improving Factual Consistency for Knowledge-Grounded Dialogue Systems via Knowledge Enhancement and Alignment

**Boyang Xue[1,*], Weichao Wang[2,*], Hongru Wang[1], Fei Mi[2], Rui Wang[3], Yasheng Wang[2], Lifeng Shang[2], Xin Jiang[2], Qun Liu[2], and Kam-Fai Wong[1,†]**

[1]The Chinese University of Hong Kong
[2]Huawei Noah's Ark Lab
[3]Harbin Institute of Technology, Shenzhen, China
{byxue, kfwong}@se.cuhk.edu.hk, wangweichao9@huawei.com

## Abstract

Pretrained language models (PLMs) based knowledge-grounded dialogue systems are prone to generate responses that are factually inconsistent with the provided knowledge source. In such inconsistent responses, the dialogue models fail to accurately express the external factual knowledge they rely upon. Inspired by previous work which identified that feed-forward networks (FFNs) within Transformers are responsible for factual knowledge expressions, we investigate two methods to efficiently improve the factual expression capability of FFNs by knowledge enhancement and alignment respectively. We first propose K-DIAL, which explicitly introduces extended FFNs in Transformers to enhance factual knowledge expressions given the specific patterns of knowledge-grounded dialogue inputs. Additionally, we apply the reinforcement learning for factual consistency (RLFC) method to implicitly adjust FFNs' expressions in responses by aligning with gold knowledge for the factual consistency preference. To comprehensively assess the factual consistency and dialogue quality of responses, we employ extensive automatic measures and human evaluations including sophisticated fine-grained NLI-based metrics. Experimental results on WoW and CMU_DoG datasets demonstrate that our methods efficiently enhance the ability of the FFN module to convey factual knowledge, validating the efficacy of improving factual consistency for knowledge-grounded dialogue systems.[1]

## 1 Introduction

Pretrained dialogue models with the assistance of external knowledge sources have demonstrated remarkable performance to generate knowledgeable

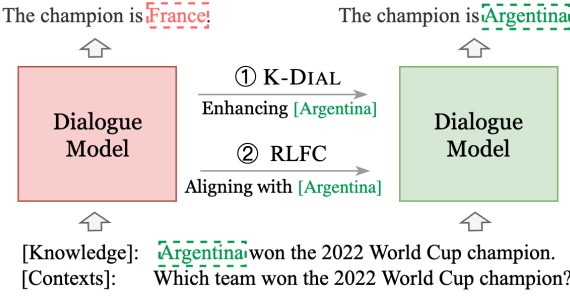

The champion is France The champion is Argentina

[Knowledge]: Argentina won the 2022 World Cup champion.
[Contexts]: Which team won the 2022 World Cup champion?

Figure 1: An illustration of enhancing the factual knowledge expression to tackle the inconsistency problem for knowledge-grounded dialogue system in this work.

and reliable responses in many conversational applications (Dinan et al., 2019; Moghe et al., 2018; Ghazvininejad et al., 2018; Gopalakrishnan et al., 2019). However, these knowledge-grounded dialogue systems (KDS) are always hampered by factual inconsistency or even "hallucination" problem (Santhanam et al., 2021; Ji et al., 2023), which has been widely investigated in many natural language generation (NLG) tasks such as abstractive summarization (Zhu et al., 2021; Nan et al., 2021; Xie et al., 2021; She et al., 2023) and machine translation (Lee et al., 2019). The factually inconsistent responses produced by dialogue models are linguistically fluent and contextually coherent but deviate from the grounding factual knowledge, as exemplified in the left hand of Figure 1, potentially leading to misinformation to the users and restricting the applicability of dialogue agents.

The factual consistency in KDS indicates "accurately portrays the input knowledge (assuming the provided knowledge is correct)" as defined in prior research (Santhanam et al., 2021). Identifying the intrinsic causes of factual inconsistency in KDS remains persistently challenging, as the generated responses are jointly affected by conversational history, grounding knowledge, and dialogue PLMs. Generally, the dialogue context and

---

*Equal contributions.
†Corresponding author.
[1]Our code has been released on https://github.com/Amour Waltz/FactDial.

grounding knowledge are assumed to be accurate and considered as ground truth, the factual inconsistency thus can be naturally attributed to the innate limitations of PLMs. The prior knowledge in PLMs learned from large-scale unlabeled corpus might be incomplete, outdated, or incorrect (Elazar et al., 2021; Cao et al., 2021) thus inevitably affecting the provided factual knowledge expressions, and consequently results in untrustworthy responses as shown in Figure 1, where the knowledge stored in the model is likely to be *"France won the World Cup champion."*. Therefore, it is essential to figure out the mechanism by which language models express factual knowledge. Previous research (Geva et al., 2021a; Dai et al., 2022a) observed that feed-forward networks (FFNs) in Transformers can be viewed as key-value memories that store and activate specific knowledge representations given certain input patterns. Accordingly, we propose two promising solutions to bolster FFNs' ability to produce factual knowledge and enhance factual consistency for KDS.

First, we propose K-DIAL, a knowledge-enhanced dialogue generation method that explicitly incorporates extended FFN modules in Transformers for knowledge enhancement, which improves the model's ability to express the gold knowledge given specific knowledge snippets and dialogue contexts. As illustrated in Figure 1, the factual knowledge *"Argentina won the 2022 World Cup champion."* with the contexts is directly used to enhance the expression of *"Argentina"* in the response. Notably, the parameters in extended FFNs are updated solely over the knowledge-related tokens occurring in both grounding knowledge and responses, ensuring the efficiency of improved factual consistency while maintaining the dialogue quality of KDS.

Second, we propose the reinforcement learning for factual consistency (RLFC) technique, which leverages alignment with the factual consistency preference to implicitly enable FFNs to express factual knowledge in responses. As shown in Figure 1, the response is aligned with factual knowledge *"Argentina won the 2022 World Cup champion."* to implicitly adjust FFNs to express accurately. The reward model is utilized for alignment which is a binary NLI model as a factual consistency classifier trained on publicly available benchmarks (Santhanam et al., 2021; Gupta et al., 2022; Dziri et al., 2021) for factual consistency evaluations of KDS.

The obtained consistency score of the reward model is utilized for RLFC training to induce factual expressions within FFNs.

To assess the factuality and conversationality of dialogue generations, we conduct a comprehensive evaluation employing both automated and human evaluations, including our carefully defined finely-grained NLI metrics based on recent human-annotated datasets released by Labadie et al. (2021); Dziri et al. (2022); Gupta et al. (2022). Significant performance improvements across the aforementioned metrics are obtained on both WoW (Dinan et al., 2019) and CMU_DoG (Zhou et al., 2018) datasets using K-DIAL and RLFC, demonstrating their efficiency in improving the expressions of factual knowledge within FFNs and mitigating the risk of factual inconsistency for KDS.

Our contributions are summarized as follows:

(1) We propose K-DIAL, which explicitly extends FFN modules in Transformers for knowledge enhancement to express factual knowledge in responses to improve factual consistency while maintaining conversational properties.

(2) We propose RLFC, which implicitly promotes FFNs' ability of factual expressions by aligning generated responses with the gold knowledge for factual consistency preference of KDS.

(3) We obtain significant improvements across a range of sophisticated automatic and human evaluation metrics, demonstrating the efficacy of our two proposed methods in achieving superior performance in terms of both the factual consistency and dialogue quality of KDS.

## 2 Methodology

In this section, we first introduce the KDS models in this work and pose the view of key-value memories of FFNs in Transformer models. We then present our knowledge-enhanced dialogue generation method K-DIAL and reinforcement learning for factual consistency (RLFC) technique respectively.

### 2.1 Knowledge-Grounded Dialogue Model

As depicted in Figure 2, the causal graph illustrates the procedure of KDS generation where response $\mathcal{Y}$ is jointly determined by dialogue history $\mathcal{X}$, retrieved knowledge $\mathcal{K}$ and PLM $\mathcal{M}$. In this study, we employ GPT2 (Radford et al., 2019) as PLMs and fine-tune these models on grounded dialogue datasets and obtain the dialogue model. The in-

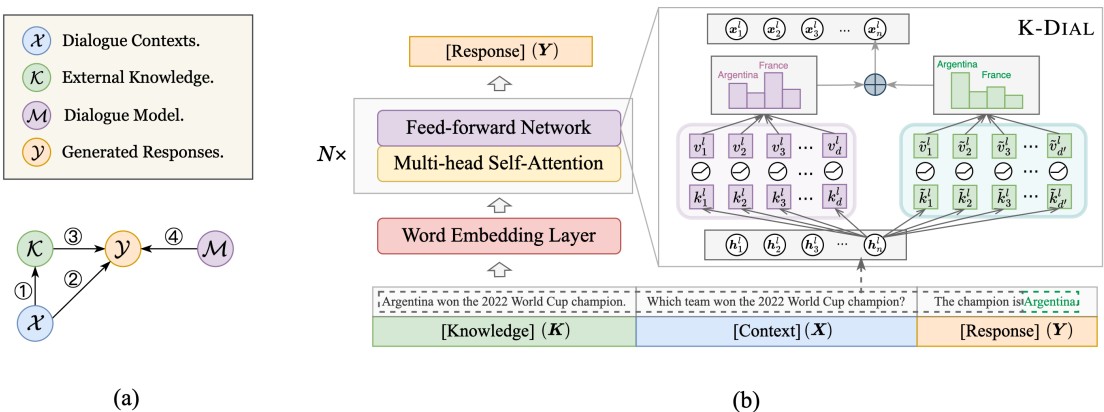

(a)                   (b)

Figure 2: An illustration of (a) a causal graph denoting the process of knowledge-grounded dialogue generations, where factual inconsistency issue in this work is attributed in $\mathcal{M}$ and procedure ④ while others are assumed correct; (b) our proposed K-DIAL framework in a dialogue model with a sample.

put to the model concatenates a piece of knowledge $K$ and a dialogue history $X$ consisting of utterances that are segmented by the speaker types <bot> and <user>. Distinct special token-type embeddings are employed to delineate each part of the input for all GPT-2 models. For simplicity, we directly leverage the gold knowledge in this work thus the input knowledge is naturally correct. The dialogue model is trained to generate the response $Y = [\boldsymbol{y}_1, \boldsymbol{y}_2, \ldots, \boldsymbol{y}_m]$ given the input via minimizing the cross-entropy loss:

$$\mathcal{L}_{\mathrm{CE}} = -\frac{1}{m} \sum_{i=1}^{m} \log p(\boldsymbol{y}_i | \boldsymbol{y}_{<i}, X, K) \quad (1)$$

### 2.2 Key-Value Memories in FFNs

Prior studies have exhibited PLMs are knowledge base (Petroni et al., 2019) and the knowledge is implicitly preserved in the parameters of FFNs in Transformers (Dai et al., 2022a). Generative PLMs, such as GPT-3 or GPT-2 (Brown et al., 2020; Radford et al., 2019), feature a deep stacking of multiple Transformer decoder blocks (Vaswani et al., 2017). As shown in Figure 2 (b), each feed-forward network (FFN) module in a Transformer block contains a two-layer linear model with an activation function between. Assume that $\boldsymbol{h}_i^l \in \mathbb{R}^{d_m}$ represents the $i$-th hidden input of the FFN module in the $l$-th Transformer layer with $d_m$-dimension word embedding . The normalized $\boldsymbol{h}_i^l$ is then fed into FFN as:

$$\mathbf{FFN}(\boldsymbol{h}_i^l) = \boldsymbol{\Theta}_{\mathrm{v}}^l \cdot \mathbf{Act}(\boldsymbol{\Theta}_{\mathrm{k}}^l \cdot \boldsymbol{h}_i^l) \quad (2)$$

where $\boldsymbol{\Theta}_{\mathrm{k}}^l \in \mathbb{R}^{d \times d_m}, \boldsymbol{\Theta}_{\mathrm{v}}^l \in \mathbb{R}^{d_m \times d}$ denote the weight matrices of the FFNs and $\mathbf{Act}(\cdot)$ is the

activation function. The bias terms are omitted.

Geva et al. (2021b) pointed that $\boldsymbol{\Theta}_{\mathrm{k}}^l$ in FFNs corresponding to keys are multiplied with $\boldsymbol{h}^l$ to yield $d$ memory coefficients. Each individual key $\boldsymbol{k}_i^l \in \boldsymbol{\Theta}_{\mathrm{k}}^l$ can capture a textual pattern across the input prefix $[\boldsymbol{h}_1^l, \cdots, \boldsymbol{h}_n^l]$ and only be triggered upon the occurrence of specific input patterns. The coefficients are then employed to compute the weighted sum with values $\boldsymbol{\Theta}_{\mathrm{v}}^l$ to induce a distribution over the vocabulary of the next token prediction. Dai et al. (2022a) further proposes the concept of knowledge neurons in FFNs that can store and activate the factual knowledge prediction. The observations provide insight to improve factual consistency for KDS by augmenting PLMs to recall and output factual knowledge in responses.

### 2.3 K-DIAL: Knowledge-Enhanced Dialogue Generations

KDS is supposed to generate more reliable and knowledgeable responses for knowledge-intensive situations leveraging the wealth of information in external knowledge. Even though gold knowledge is provided, the models still encounter challenges related to fictional expressions of gold knowledge they rely on, and resulting in factual inconsistency, for example, manifesting in the responses that are factually incorrect or uninformative. As shown in Figure 2 (a) where $\mathcal{X}$ and $\mathcal{K}$ are considered always correct, we can naturally infer that the inconsistency arises from $\mathcal{M}$.

As the knowledge in PLMs inevitably contains inaccurate, outdated, incomplete redundant information (Elazar et al., 2021; Cao et al., 2021), which may influence factual knowledge predictions of

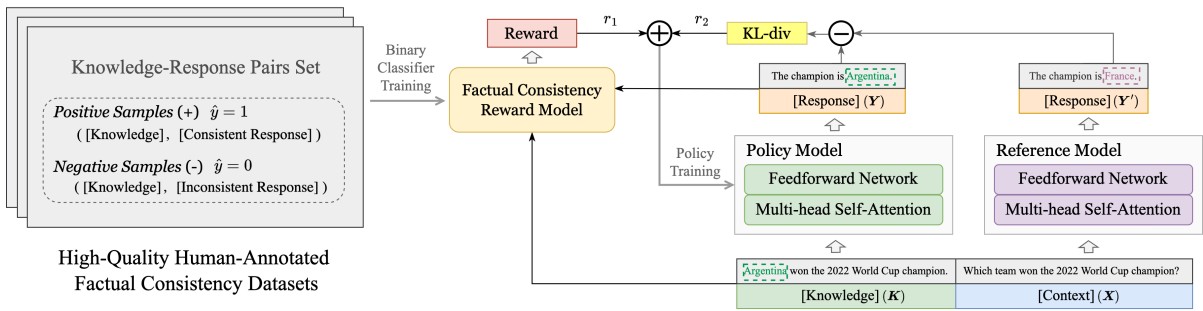

Figure 3: Reward model training and workflow of RLFC training for KDS.

FFNs. Factual knowledge, or world knowledge, is generally represented among entities in languages (i.e., dates or locations) (Roberts et al., 2020). As illustrated in Figure 2(b), we propose K-DIAL, which directly extended an additional FFN module with $d'$ key-value pairs and further concatenated with the original FFNs of PLMs, to maximize the activation of each entity token $y_k$ over a certain knowledge-grounded dialogue input pattern of the sequence $[\boldsymbol{K}, \boldsymbol{X}, \boldsymbol{y}_{<k}]$. The loss function of K-DIAL framework is formulated as follows:

$$\mathcal{L}_{\text{KCE}} = -\frac{1}{n'} \sum_{i=1}^{m} \mathbb{1}_{\tilde{\boldsymbol{y}}_k}(\boldsymbol{y}_i) \log p(\boldsymbol{y}_i|\boldsymbol{y}_{<i}, \boldsymbol{X}, \boldsymbol{K})$$

(3)

where $n'$ is the number of entities in $\boldsymbol{Y}$ and $\mathbb{1}_{\tilde{\boldsymbol{y}}_k}(\boldsymbol{y}_i)$ denotes whether $\boldsymbol{y}_i$ belongs to the entity set $\tilde{\boldsymbol{y}}_k$.

The training process of K-DIAL framework is specified in two steps. First, all the parameters of the original PLM are frozen, and the loss $\mathcal{L}_{\text{KCE}}$ is only calculated over the parameters of the extended FFNs. Afterward, we further adapt the knowledge-enhanced model on the KDS datasets using supervised fine-tuning of Equation (1) while keeping the parameters in extended FFNs fixed. The word embedding dimension and hidden size of extended FFN module are set equal to the corresponding Transformer FFN layers. Note that K-DIAL framework is only applied on the top 3 layers of the model in our experiments.

As illustrated in Figure 2, the extended FFNs are supposed to predict *Argentina* as the next token given the specific knowledge and context. The K-DIAL framework takes advantage of FFNs' ability to learn the complex dependency between the knowledge snippet and dialogue via activating related entity tokens. In this way, factual consistent entity words can be triggered in the response. The ability of PLMs to express factual knowledge has been improved while maintaining the general language ability.

## 2.4 RLFC: Reinforcement Learning for Factual Consistency

For KDS, we prefer knowledgeable responses that are faithful to the gold knowledge. However, since PLMs are trained on the massive unlabeled corpus, KDS models do not inherently prioritize following the preferences to constantly output factual knowledge and consistent responses. Aligning with the factual consistency preference can implicitly encourage FFNs of Transformers to convey factual knowledge and ultimately reinforce the factual consistency of responses. Inspired by the recent progress of reinforcement learning from human feedback (RLHF) technique to align human preferences (Ouyang et al., 2022; Ziegler et al., 2019; Christiano et al., 2017) like mitigating toxicity (Faal et al., 2023), we regard factual consistency as one type of preferences and thus propose reinforcement learning for factual consistency (RLFC) method in this work.

Specifically, we first design a reward model using a factual consistency classifier. There are some recent publicly human-annotated benchmarks and datasets containing information on the preference for factual consistency (Santhanam et al., 2021; Gupta et al., 2022; Dziri et al., 2021), where the similar definitions to factual consistency as "attributed" and "supported" are used in KDS indicating whether the response utilizes and follows the provided knowledge. We thus take advantage of these well-aligned data to train a binary NLI model, serving as a reward model to provide informative reward signals for RL training. The reward model $R(\cdot)$ is optimized using the following binary cross-entropy loss function:

$$\mathcal{L}_{\text{BCE}} = -\frac{1}{n} \sum_{i=1}^{n} (\hat{\boldsymbol{y}}^{(i)} \log R(\boldsymbol{K}^{(i)}, \boldsymbol{Y}^{(i)})$$
$$+ (1 - \hat{\boldsymbol{y}}^{(i)}) \log (1 - R(\boldsymbol{K}^{(i)}, \boldsymbol{Y}^{(i)}))) \tag{4}$$

where the knowledge-response pair $(\boldsymbol{K}^{(i)}, \boldsymbol{Y}^{(i)})$ is the input to the reward model and $\hat{\boldsymbol{y}}^{(i)}$ is the label.

As illustrated in Figure 3, the dialogue model to be optimized for RLFC training is used as the policy model. The response $\boldsymbol{Y}$ generated by the policy model and gold knowledge snippet $\boldsymbol{K}$ are fed into the reward model to obtain the consistency reward score $r_1$ as $r_1 = R(\boldsymbol{K}, \boldsymbol{Y})$ which is mainly used to align the preferences for FFNs' factual expression. The reward model will return a higher score for the factually consistent pairs to facilitate the factual expressions of the policy model. Furthermore, a reference model generating a response $\boldsymbol{Y}'$ is also introduced, which is usually the dialogue model before RLFC training. The KL divergence $r_2 = \text{KL}[\boldsymbol{Y}||\boldsymbol{Y}']$ between the outputs of the reference model and the policy model is used as an extra reward signal to make sure the generated responses don't diverge too far from the originals. The optimization objective $r = r_1 + r_2$ is utilized for RLFC training via the Proximal Policy Optimization (PPO) algorithm (Schulman et al., 2017).

## 3 Experimental Setup

### 3.1 Datasets

**WoW** The Wizard of Wikipedia (WoW) [2] is a large-scale multi-turn knowledge-grounded dialogue dataset (Dinan et al., 2019) collected through human-human crowd-worker chats, where a "wizard" as the <bot> can respond to an "apprentice" as a <user> in a knowledgeable way given evidence from external Wikipedia documents. We only focus on modeling the responses by the "wizard" provided the selected gold-label evidence and the previous dialogue contexts.

**CMU_DoG** The CMU Document Grounded Conversations Dataset (CMU_DoG) [3] (Zhou et al., 2018) refers to a collection of conversations that encompass two users discussing various famous movies given related Wikipedia documents. Utterances by the user who can access the movie evidence in the documents are treated as <bot>

responses for dialogue modeling. Note that the initial configuration of CMU_DoG entails the provision of a gold knowledge paragraph to the models alongside the dialogue. In this work, we split these knowledge documents into sentence pieces and select the most relevant one as the grounding knowledge, preserving the average token number of knowledge snippets comparable to those on WoW.

More data processing details can be referred to in Appendix A.

### 3.2 Implementation Details

For the dialogue generation models, we leverage GPT2 series (GPT2-Medium(M), GPT2-Large(L), GPT2-XL) models (Radford et al., 2019) implemented using HuggingFace library (Wolf et al., 2020) [4] based on PyTorch (Paszke et al., 2017). All the PLMs are further fine-tuned on the above WoW and CMU_DoG dialogue datasets by minimizing the cross-entropy loss in Equation (1). ADAM parameter update is used in a mini-batch mode for all models. In the decoding stage, we use the beam search algorithm and set the number of beams $n = 5$. During K-DIAL training, all the knowledge entities in gold knowledge and responses are recognized using spaCy[5]. The RLFC is implemented by trl[6] in this work, and all the hyperparameters related to PPO algorithm are default values by the trl PPOConfig recipe [7] except the epoch, learning rate and batch size. The reward model is obtained by training a BERT-Large (Devlin et al., 2019) based NLI model as a factual consistency classifier trained on three public, high-quality human-annotated benchmarks and datasets Santhanam et al. (2021)[8], Dziri et al. (2021) [9], Gupta et al. (2022) [10].

Further model setting information can be found in Appendix B.

### 3.3 Metrics

We exhibit a range of comprehensive metrics to gauge the factuality and conversational ability of

[2]https://parl.ai/projects/wizard_of_wikipedia/
[3]https://github.com/festvox/datasets-CMU_DoG
[4]https://huggingface.co/gpt2
[5]https://spacy.io/
[6]https://github.com/lvwerra/trl
[7]https://github.com/huggingface/trl/blob/main/trl/trainer/ppo_config.py
[8]https://github.com/alexa/factual-consistency-analysis-of-dialogs
[9]https://github.com/google/BEGIN-dataset
[10]https://github.com/salesforce/DialFact

KDS, entailing both a series of automated techniques as well as human evaluations.

**Lexical and Semantic Metrics** In this work, we adopted token-level F1 uni-gram overlap, BLEU, and ROUGE-L metrics to measure the lexical similarity for dialogue quality evaluation between generated and ground-truth responses. Additionally, Knowledge F1 (KF1) (Shuster et al., 2021) and BERTScore (Zhang* et al., 2020) (**BERT.**) are used to capture such lexical overlap and semantic similarity of response and grounding knowledge.

**Fine-Grained NLI-based Metrics** NLI-based metrics are more robust and widely used to detect factual inconsistency or hallucinations in knowledge-intensive NLP tasks (Dušek and Kasner, 2020; Mishra et al., 2021; Falke et al., 2019; Welleck et al., 2019; Chen et al., 2023). Therefore, we developed a synthetic dataset for a BERT-based (Devlin et al., 2019) NLI model pre-training. The synthetic dataset adopts factual consistent samples that are derived from the ground-truth response and gold knowledge pairs in WoW. Inconsistent responses are generated by random pairing, negation, and entity swapping (Kryscinski et al., 2020a; Gupta et al., 2022).

Following (Santhanam et al., 2021), we develope three metrics to finely-grained evaluate factual consistency and fine-tune the pre-trained NLI model on the datasets released by (Santhanam et al., 2021; Gupta et al., 2022; Dziri et al., 2021), which are also used for reward model training of RLFC. The three fine-grained NLI metrics are designed to inspect 1) *Verification* (**Verif.**): whether a response contains verifiable information; 2) *Hallucination* (**Hallu.**): whether a response does **NOT** comprises hallucinated content; and 3) *Factual Correctness* (**Fact.**): whether a response is factually consistent with grounding knowledge.

Although there may be slight variations across the definitions in the aforementioned benchmarks, their shared objective is to enhance the faithfulness and reliability of responses to the provided gold knowledge. The data processing and alignment details are presented in Appendix C.

$Q^2$ **Metric** Honovich et al. (2021) proposed $Q^2$ metric [11] employed a question generation system, a question answering system, and an NLI model to find the corresponding answer span in the knowl-

---

[11] https://github.com/orhonovich/q-squared

edge that the response should be grounded to evaluate the factual consistency of KDS.

**Human Evaluation** We exhibit human evaluations as a means of assessing performance across various dimensions of generation quality. Annotators were requested to answer: 1) *whether the response is fluent and understandable* (**Flue.**) and 2) *whether the response is contextually coherent given previous conversations* (**Cohe.**) and 3) *whether the response is factually correct* (**Fact.**).

All the annotators were asked to rate each quality on a two-level Likert scale from 0 (*Not Flunet, Not Coherent, Inconsistent*) to 1 (*Flunet, Coherent, Consistent*) to evaluate the fluency, coherence, and factual consistency of generated responses. The averaged results by the human evaluation scores are reported.

## 4 Results and Analysis

### 4.1 Main Experiments of K-DIAL and RLFC

**Results of Automatic Evaluations** In Table 1, we present the experimental results of various GPT2-based dialogue PLMs using K-DIAL and RLFC on WoW and CMU_DoG test sets. Several trends can be found below:

1) **The effects of K-DIAL:** GPT2 series models using K-DIAL outperform all standard dialogue models in both factual consistency and dialogue quality for KDS on both WoW and CMU_DoG test sets. Significant factual consistency improvements of GPT2-L+K-DIAL in Fact. and $Q^2$ in terms of 5.36% and 7.63% absolutely over GPT2-L on WoW indicate that K-DIAL effectively enhances factual expressions. On CMU_DoG, supreme factual consistency improvements of 3.84% and 4.11% absolutely on Fact. and $Q^2$ are obtained on GPT2-M after using K-DIAL.

Improvements in the KF1 measure suggest that the responses equipped by K-DIAL are more knowledgeable and faithful to the supported knowledge. The performance improvements obtained on the conversationality metrics of BLEU, F1, and ROUGE-L present that through enhancing factual expressions for responses, the dialogue quality can be also marginally improved.

2) **The effects of RLFC:** Comparable performance improvements are also acquired using RLFC on GPT2 dialogue models, demonstrating that RLFC can proficiently improve the factual consistency of KDS on both WoW and CMU_DoG,

| Model | Dataset | KF1 | BERT. | Verif. | Hallu. | Fact. | $Q^2$ | BLEU | F1 | ROUGE |
|---|---|---|---|---|---|---|---|---|---|---|
| GPT2-M | | 46.51 | 38.25 | 13.67 | 10.94 | 5.74 | 55.55 | 27.09 | 60.83 | 7.83 |
| + K-DIAL | | 48.36 | 41.97 | 16.67 | 11.54 | 9.36 | 62.47 | 28.33 | 61.48 | 8.69 |
| + RLFC | | 47.23 | 40.34 | 18.84 | 11.15 | 8.44 | 59.14 | 25.43 | 59.26 | 6.24 |
| + RLFC + K-DIAL | | 50.27 | 41.65 | 18.74 | 12.37 | 11.56 | 63.26 | 27.45 | 61.34 | 7.98 |
| GPT2-L | | 68.46 | 47.44 | 43.96 | 32.35 | 40.19 | 76.08 | 53.32 | 76.18 | 30.82 |
| + K-DIAL | WoW | 70.59 | 50.73 | 45.38 | 34.12 | 45.55 | 83.71 | 55.78 | 77.64 | 32.45 |
| + RLFC | | 70.44 | 52.27 | 47.73 | 35.16 | 44.41 | 80.25 | 55.42 | 75.25 | 31.24 |
| + RLFC + K-DIAL | | 72.38 | 53.25 | 48.81 | 37.98 | 46.37 | 82.72 | 56.78 | 77.57 | 33.34 |
| GPT2-XL | | 73.67 | 51.40 | 50.15 | 34.44 | 48.40 | 79.38 | 54.45 | 79.72 | 36.90 |
| + K-DIAL | | 75.48 | 53.38 | 50.07 | 36.45 | 49.12 | 83.38 | 53.96 | 80.23 | 36.84 |
| + RLFC | | 75.21 | 52.26 | 51.33 | 36.17 | 49.31 | 82.01 | 54.60 | 80.11 | 37.02 |
| + RLFC + K-DIAL | | 76.35 | 53.60 | 50.94 | 37.08 | 50.13 | 83.87 | 54.77 | 79.99 | 37.19 |
| GPT2-M | | 26.13 | 32.35 | 10.27 | 7.54 | 4.26 | 37.51 | 36.68 | 61.33 | 19.12 |
| + K-DIAL | | 29.21 | 37.44 | 12.08 | 8.94 | 8.10 | 41.62 | 37.64 | 62.19 | 19.39 |
| + RLFC | | 30.54 | 35.20 | 12.96 | 8.21 | 7.27 | 40.43 | 38.24 | 62.13 | 20.13 |
| + RLFC + K-DIAL | | 31.66 | 38.12 | 14.09 | 9.89 | 9.23 | 42.36 | 38.56 | 62.49 | 21.07 |
| GPT2-L | | 51.35 | 39.42 | 31.54 | 27.65 | 23.38 | 57.82 | 45.17 | 68.55 | 31.27 |
| + K-DIAL | CMU_DoG | 53.16 | 42.23 | 33.64 | 30.06 | 25.45 | 59.40 | 45.74 | 69.16 | 32.57 |
| + RLFC | | 52.87 | 42.36 | 33.95 | 28.71 | 25.31 | 59.12 | 46.72 | 69.88 | 33.37 |
| + RLFC + K-DIAL | | 54.02 | 44.89 | 34.70 | 31.43 | 28.67 | 61.34 | 46.65 | 70.18 | 34.09 |
| GPT2-XL | | 65.14 | 45.09 | 42.88 | 35.17 | 33.97 | 66.13 | 48.25 | 75.39 | 34.35 |
| + K-DIAL | | 66.48 | 45.26 | 44.08 | 37.24 | 34.53 | 68.50 | 48.79 | 75.22 | 35.52 |
| + RLFC | | 66.11 | 51.33 | 44.06 | 37.16 | 34.83 | 69.34 | 50.02 | 75.64 | 35.86 |
| + RLFC + K-DIAL | | 67.41 | 46.21 | 44.97 | 38.03 | 35.05 | 70.14 | 51.14 | 75.49 | 36.10 |

Table 1: Experiments of GPT2 series models fine-tuned on KDS datasets employed with K-DIAL and RLFC methods on WoW and CMU_DoG test set.

where performance improvements of 3.77% in Verif. on WoW and 2.41% on CMU_DoG are obtained by GPT2-L+RLFC over GPT2-L models.

RLFC performs better on Verif. measure over standard baseline models than K-DIAL, suggesting that RLFC is better at promoting the model's ability to generate verifiable responses by aligning with factual knowledge. The side-effect of degradation on dialogue quality metrics implies that applying RLFC uniquely cannot effectively maintain the original conversationality of the standard GPT2 dialogue model.

3) **The effects of RLFC+K-DIAL:** The optimal training strategy to obtain the final models is specialized in two stages. We first train the GPT2 models using RLFC and then apply K-DIAL on the obtained model. The supreme performance improvements are obtained on the setting of GPT2-L using the combination of RLFC and K-DIAL methods in $Q^2$ and Fact. in respective of 6.18% and 8.64% absolutely over standard GPT2-L dialogue model on WoW. The combination of RLFC and K-DIAL can implicitly and explicitly improve the models' ability to express factual knowledge as a complementary, demonstrating the best performance in respect of factual consistency and dialogue quality.

4) **The effects of model size:** We observe that better performance improvements are attained on GPT2-M and GPT2-L models using either K-DIAL or RLFC than on larger-scale GPT2-XL models on both WoW and CMU_DoG test sets, as the GPT2-XL models finetuned on KDS datasets are more robust to generate factual consistent contents.

| Model | Dataset | Flue. | Cohe. | Fact. |
|---|---|---|---|---|
| GPT2-L | | 1.00 | 0.88 | 0.65 |
| + K-DIAL | WoW | 1.00 | 0.90 | 0.68 |
| + RLFC | | 1.00 | 0.91 | 0.69 |
| + RLFC + K-DIAL | | 1.00 | 0.91 | 0.74 |
| GPT2-L | | 1.00 | 0.86 | 0.69 |
| + K-DIAL | CMU_DoG | 1.00 | 0.86 | 0.73 |
| + RLFC | | 1.00 | 0.87 | 0.74 |
| + RLFC + K-DIAL | | 1.00 | 0.86 | 0.76 |

Table 2: Human evaluation results of GPT2-L model using respective and combination of proposed K-DIAL and RLFC on 100 samples selected from WoW and CMU_DoG test sets respectively.

**Results of Human Evaluations**    To accurately gauge the performance of the proposed methods, we perform human evaluations in Table 2. We select a subset of examples from the WoW and CMU_DoG test sets, using 100 examples from each dataset per model variant with 3 human raters. Results indicate that responses generated by all the models are fluent and coherent to appropriately make sense in engaging the context of the conversations. Furthermore, the assessment of factual consistency by human evaluators demonstrates a

| Model | Dataset | # Para | KF1 | BERT. | Verif. | Hallu. | Fact. | $Q^2$ | BLEU | F1 | R.L. |
|---|---|---|---|---|---|---|---|---|---|---|---|
| GPT2-M | | 355M | 46.51 | 38.25 | 13.67 | 10.94 | 5.74 | 55.55 | 27.09 | 60.83 | 7.83 |
| + K-ADAPTER | | 377M | 46.96 | 39.32 | 14.66 | 10.35 | 7.64 | 60.36 | 26.03 | 59.17 | 7.06 |
| + K-Former | WoW | 361M | 47.43 | 39.91 | 15.51 | 11.50 | 8.37 | 57.87 | 26.28 | 59.08 | 8.14 |
| + NKB | | 361M | 46.60 | 38.74 | 13.49 | 9.51 | 6.35 | 58.88 | 25.30 | 60.55 | 6.84 |
| + K-DIAL (ours) | | 361M | 48.36 | 41.97 | 16.67 | 11.54 | 9.36 | 64.47 | 28.33 | 61.48 | 8.69 |
| + K-DIAL-$\alpha$ (ours) | | 361M | 49.03 | 41.66 | 16.50 | 11.23 | 9.48 | 62.34 | 29.36 | 61.06 | 8.40 |
| GPT2-M | | 355M | 26.13 | 32.35 | 10.27 | 7.54 | 4.26 | 37.51 | 36.68 | 61.33 | 19.12 |
| + K-ADAPTER | | 377M | 27.69 | 34.44 | 11.32 | 8.63 | 6.64 | 39.26 | 37.26 | 61.19 | 18.84 |
| + K-Former | CMU_DoG | 361M | 28.90 | 35.89 | 12.61 | 9.06 | 7.37 | 40.35 | 37.18 | 61.48 | 19.24 |
| + NKB | | 361M | 27.63 | 32.97 | 10.59 | 9.51 | 5.23 | 36.86 | 36.45 | 61.30 | 18.55 |
| + K-DIAL (ours) | | 361M | 29.21 | 37.44 | 12.08 | 8.94 | 8.10 | 41.62 | 37.64 | 62.19 | 19.39 |
| + K-DIAL-$\alpha$ (ours) | | 361M | 28.83 | 37.25 | 12.20 | 9.23 | 7.68 | 40.28 | 37.77 | 62.10 | 20.84 |

Table 3: Experiments and parameter use of several knowledge enhancement baseline and K-DIAL variants comparisons on GPT2-M on WoW and CMU_DoG test set.

| Grounding Documents | Dialogue History | Generated Responses |
|---|---|---|
| 📖 1980s in Music

As disco fell out of fashion in the decade's early years, genres such as post-disco, Italo disco, Euro disco, and dance-pop became more popular in the 1980s. | 🙂 `<user>` Music in the 1980s was so much fun from what I have heard?

🤖 `<bot>` Yeah the 1980's was the emergence of dance music and new wave music.

🙂 `<user>` What exactly was the type of music coming out in the 1980s? Pop? | 🤖 `<bot>` **Baseline:** The popular type of music coming out in the 1980s was disco.

🤖 `<bot>` **K-DIAL:** Dance-pop, post disco, Italo disco, and Euro disco became more popular in the 1980s.

🤖 `<bot>` **RLFC:** As I know prior to the 1980s, disco was the most popular fashion of music. |

Figure 4: An example of a conversation on WoW valid set before and after using K-DIAL and RLFC on the dialogue model (denoted as `<bot>`). Incongruous content is highlighted in Red, while the counterpart gold knowledge in the Wikipedia document is in Blue.

strong correlation with the Fact. and $Q^2$ metrics in Table 1, which confirms that our proposed methods exactly improve factual consistency for KDS. The raters' agreements for each quality are measured separately using Fleiss' Kappa of statsmodels [12]. All the results (Flue.:0.99, Cohe.:0.95, Fact.:0.77 respectively) demonstrate substantial and almost perfect agreement levels.

### 4.2 Baseline Methods and K-DIAL Variants Comparisons

**Experiments of Baseline Comparisons** To the best of our knowledge, this work is the first to propose to improve factual consistency for KDS. Previous related works that investigate the factual consistency of KDS only focus on evaluation methods (Honovich et al., 2021) or datasets (Labadie et al., 2021; Dziri et al., 2022; Gupta et al., 2022). Therefore, there is no direct improvement method to be compared.

Nevertheless, for the knowledge-enhancing method K-DIAL, we still carry out experiments on several knowledge injection and enhancement methods for NLG tasks, including K-Adapter (Wang et al., 2021), Kformer (Yao et al., 2022), and

[12]https://github.com/statsmodels

eural Knowledge Bank (NKB) (Dai et al., 2022b), which first integrate substantial exogenous knowledge and are further adapted on KDS tasks as baselines presented in Table 3. Experimental results on GPT2-M dialogue models generally show that all the knowledge injection methods can marginally improve the factual consistency but slightly degrade the dialogue quality on BLEU and ROUGE-L. K-DIAL outperforms the three baseline knowledge injection methods in both factuality and conversationality, demonstrating superior performance to improve factual consistency without sacrificing the dialogue qualities for KDS tasks.

More details regarding the baseline configurations and implementations are available in Appendix D.

**Experiments of K-DIAL Variants** We also conduct variant comparison experiments of a K-DIAL-$\alpha$ which updates the extended FFN modules using $\mathcal{L}_{CE}$ as Equation (1) and calculate the loss on all tokens rather than $\mathcal{L}_{KCE}$ of Equation (3) for just knowledge entities. Experimental results suggest that optimizing K-DIAL by either $\mathcal{L}_{CE}$ or $\mathcal{L}_{KCE}$ has comparable performance, as the knowledge information is mainly represented by the knowledge entities and learned by the FFN modules. For ef-

ficiency, we adopt only updating extended FFN parameters on knowledge entities for the K-DIAL method.

## 4.3 Case Analysis of K-DIAL and RLFC

We further present a representative case in Figure 4 to analyze the practical performance of proposed K-DIAL and RLFC methods in comparison with the standard GPT2-L model respectively. The following trends are found:

1) Both K-DIAL and RLFC can effectively correct the inconsistent response generated by the standard GPT2-L dialogue model that contradicts the factual knowledge in the Wikipedia document, demonstrating their efficacy in improving factual consistency for KDS.

2) The K-DIAL method is more likely to learn the important knowledge snippet and directly express it in responses, which is achieved by extended FFNs' explicit ability to enhance factual knowledge expressions.

3) RLFC implicitly aligns the FFNs' expressions with external gold knowledge for the factual consistency preference, which is more semantically natural and human-like than K-DIAL.

## 5 Related Works

**Factual Consistency in NLG** The issue of factual inconsistency in NLG tasks has attracted increasing attention in many fields such as abstractive summarization, with studies focusing on both improving and evaluating the factual consistency (Kryscinski et al., 2020b; Maynez et al., 2020; Xie et al., 2021; Zhu et al., 2021; Nan et al., 2021). Related works were also conducted on data-to-text generation (Dušek and Kasner, 2020; Thomson and Reiter, 2020). In the context of dialogue systems, Dziri et al. (2021); Gupta et al. (2022) introduced the benchmarks for measuring the attribution and fact-checking of dialogue generations with grounding knowledge. In the context of dialogue systems, Rashkin et al. (2021) added controllable tokens on the input of the dialogue model to generate responses that are more faithful to the source knowledge. Shuster et al. (2021) investigated the Retrieval-Augmented Generation (RAG) approach to reduce knowledge hallucination in conversational agents. Peng et al. (2023) introduced LLM-AUGMENT, a framework for augmenting black-box LLMs with external knowledge and automated feedback to reduce hallucinations.

**Enhancing Knowledge in PLMs** Previous works have explored ways to incorporate external knowledge into pre-trained language models (PLMs). ERNIE (Zhang et al., 2019) and Know-BERT (Zhang et al., 2019; Peters et al., 2019) enhance the word representations by incorporating external knowledge graphs. K-ADAPTER introduces two adapters to inject factual and linguistic knowledge into PLM respectively (Wang et al., 2021). Inspired by Dai et al. (2022a), recent works focused to add extended FFNs in Transformer-like K-former (Yao et al., 2022) or Neural Knowledge Bank (NKB) (Dai et al., 2022b) to inject and update extra knowledge while keeping the original model parameters frozen. Dong et al. (2022) investigated to detect the incorrect knowledge stored in PLMs and proposed CALINET for knowledge calibration.

**Reinforecment Learning in NLG** With growing interest in RL technique, learning enhanced LMs from human feedback has been explored in Ouyang et al. (2022); Bahdanau et al. (2019); Ramamurthy et al. (2022). Language models are optimized to align with human preferences such as harmless and non-toxic outputs (Faal et al., 2022; Bai et al., 2022), which means following human instructions better. RLHF technique was also widely applied in downstream tasks such as dialogue system (Jaques et al., 2020; Lu et al., 2022; Kwan et al., 2023; Wang et al., 2022) and abstractive summarization (Böhm et al., 2019; Wu et al., 2021).

## 6 Conclusion

In this work, we investigate the inadequacy of KDS that often produces factually inconsistent responses unsupported by grounding knowledge. We propose two strategies to tackle this issue. K-DIAL introduces extended FFN modules to explicitly enhance factual knowledge expressions given specific input patterns of the knowledge snippets and dialogue contexts. RLFC technique is used to implicitly adjust FFNs to augment factual expressions in responses by aligning with the gold knowledge for factual consistency preferences. Experiments present that both K-DIAL and RLFC can promote the knowledgeability, factual consistency and conversational ability of responses, demonstrating the efficacy of our proposed methods to improve the ability of FFNs to express factual knowledge to generate more informative and reliable responses in dialogue applications.

## Limitations

The limitations of this work are summarized below:

1) As shown in Figure 2 (a) and described before, this paper assumes that factual inconsistency comes along with dialogue model $\mathcal{M}$ and procedure ④ in Figure 2 (a), which deviated from the reality that the knowledge retrieval process ① and hallucinations in knowledge $\mathcal{K}$ and contexts $\mathcal{X}$ are not always correct. A more challenging problem lies in locating the inconsistency cause of generation processes. In future work, we will make a systematic investigation of the factual inconsistency and hallucination problems in KDS.

2) Recently, large-scale language models (LLMs) such as GPT3 and ChatGPT have demonstrated state-of-the-art performance across a range of NLP tasks. This work was only conducted on the GPT2 series PLMs with a maximum of 1.26B parameters, which is extremely small in comparison with such LLMs containing hundreds of billions of parameters. However, since the proposed methods involve plenty of model parameter updating, it is difficult to employ on LLMs due to the limitations of GPU resources in the initial work. Next, we will continue to explore the transferability of the framework using the parameter-efficient method to the employment of open-source LLMs.

## Acknowledgements

This research work is partially supported by CUHK direct grant no. 4055209.

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

## A    Dataset Details

**WoW**    As listed below in Table 4, the WoW dataset contains 18,430, 1,948, and 1,933 conversations in Train/Valid/Test sets respectively. Both "seen" and "unseen" topic portions of test sets have been merged. Each conversation data spans 4-5 turns of utterances between Wizard and Apprentice. In each turn, the response by the Wizard is grounded on gold knowledge of one or two checked sentences in the Wikipedia documents as evidentiary support.

**CMU_DoG**    The CMU_DoG (Zhou et al., 2018) dataset contains 3,373, 229, 619 conversations with an average of 21.43 turns per conversation. In CMU_DoG, the grounding knowledge is at the document level, which is a more difficult (but realistic) setting rather than WoW.

The number of processed training samples are presented in 4 below.

|  | WoW | | CMU_DoG | |
|---|---|---|---|---|
|  | # Conv. | # Samp. | # Conv. | # Samp. |
| Train | 18,430 | 73,571 | 3,373 | 78,406 |
| Valid | 1,948 | 7,903 | 229 | 5,474 |
| Test | 1,933 | 7,844 | 619 | 14,700 |

Table 4: Data statistics of two datasets.

## B    Model Details

The hyper-parameters of GPT2 series models on both WoW and CMU_DoG tasks are listed in Table 5 including training epochs, batch size, learning rate, warm-up steps, and maximum sequence length.

## C    Metrics Details

**NLI-based Metrics**    For our annotation of factual consistency, we categorize responses into three types as shown in Figure 6: Non-verifiable Response does not include any information that needs to be verified and cannot be evaluated as consistent or not consistent. A factually consistent response is informative and highly relevant to the provided knowledge. Hallucinated responses may not be always consistent with the knowledge but could still be correct. Despite the exorbitant expenses associated with human annotations, there are still publicly precious gold-label datasets. Following Santhanam et al. (2021), we defined fine-grained metrics on factual consistency evaluation

with respect to *Verification*, *Factual Consistency*, and *Hallucination* as exemplified in Figure 6. Similar taxonomy is also adopted in Gupta et al. (2022) (*Generic/Attributable/Not Attributable*) and Dziri et al. (2021) (*Verification/Supported/Refuted/Not Enough Information* (NEI)).

## D    Baseline Method Details

**K-Adapter**    Wang et al. (2021) proposed K-ADAPTER, a neural adapter architecture specifying one kind of knowledge (e.g. *Factual Knowledge* or *Liguistic Knowledge*), as plug-in connections into different Transformer layers of PLMs. Following Wang et al. (2021), we set each adapter model consisting of three adapter layers plug-in among the highest, middle, and lowest Transformer layers of PLMs (e.g. for the 36-layer GPT2-Large, adapter layers plug-in is configured as {1,18,36}), and parameters are not shared across different adapter layers. Each adapter layer comprises two Transformer layers and two projection layers illustrated in Figure 5. The Transformer block of the adapter layer has been established to be of equal size to that of the PLMs. Additionally, the hidden dimensions of the down-projection and up-projection layers have been set to correspond to the word embedding and hidden dimension of the PLMs in different scales respectively.

**Kformer**    Kformer (Yao et al., 2022) is a knowledge fusion model that converts the knowledge into dense embedding vectors and then injects them into the parameters of expanded FFNs of the Transformer layers followed by Dai et al. (2022a). In accordance with (Yao et al., 2022), we encode the external knowledge via an embedding layer which is initialized as the same word embedding matrix of GPTs. Then we map the obtained knowledge representations into the corresponding vector space of FFN weights. Only the top 3 layers of all PLMs were adopted for knowledge enhancement.

**Neural Knowledge Bank**    Neural Knowledge Bank (Dai et al., 2022b) have put forth the Neural Knowledge Bank (NKB) which is an extended FFN module as the memory slots for knowledge infusion using Salient Span Masking (SSM) (Guu et al., 2020) strategy to preserve the general language modeling competency. The NKB architectures are expanded onto the top three FFN layers of GPTs. The quantity of supplementary memory slots is established to match the dimension of intermediate

| Model | Dataset | Epoch | Batch | L.R. | Warm. | Seq. Len. |
|---|---|---|---|---|---|---|
| GPT2-M | | 4 | 16 | 6e-5 | 2k | 256 |
| GPT2-M+K-DIAL (FFNs) | | 2 | 32 | 6e-5 | 2k | 256 |
| GPT2-M+K-DIAL (GPT2 Model) | | 3 | 16 | 6e-5 | 2k | 256 |
| GPT2-M+RLFC | | 3 | 16 | 1e-5 | - | 256 |
| GPT2-L | | 4 | 8 | 6e-5 | 4k | 256 |
| GPT2-L+K-DIAL (FFNs) | | 2 | 24 | 6e-5 | 4k | 256 |
| GPT2-L+K-DIAL (GPT2 Model) | WoW | 3 | 8 | 6e-5 | 4k | 256 |
| GPT2-L+RLFC | | 3 | 8 | 1e-5 | - | 256 |
| GPT2-XL | | 4 | 4 | 6e-5 | 8k | 256 |
| GPT2-XL+K-DIAL (FFNs) | | 2 | 16 | 6e-5 | 8k | 256 |
| GPT2-XL+K-DIAL (GPT2 Model) | | 3 | 4 | 6e-5 | 8k | 256 |
| GPT2-XL+RLFC | | 3 | 4 | 1e-5 | - | 256 |
| GPT2-M | | 4 | 16 | 1e-5 | 4k | 256 |
| GPT2-M+K-DIAL (FFNs) | | 2 | 32 | 1e-5 | 4k | 256 |
| GPT2-M+K-DIAL (GPT2 Model) | | 3 | 16 | 1e-5 | 4k | 256 |
| GPT2-M+RLFC | | 3 | 16 | 1e-5 | - | 256 |
| GPT2-L | | 4 | 8 | 1e-5 | 8k | 256 |
| GPT2-L+K-DIAL (FFNs) | | 2 | 24 | 1e-5 | 8k | 256 |
| GPT2-L+K-DIAL (GPT2 Model) | CMU_DoG | 3 | 8 | 1e-5 | 8k | 256 |
| GPT2-L+RL | | 3 | 8 | 1e-5 | - | 256 |
| GPT2-XL | | 4 | 4 | 1e-5 | 8k | 256 |
| GPT2-XL+K-DIAL (FFNs) | | 2 | 16 | 1e-5 | 8k | 256 |
| GPT2-XL+K-DIAL (GPT2 Model) | | 3 | 4 | 1e-5 | 8k | 256 |
| GPT2-XL+RLFC | | 3 | 4 | 1e-5 | - | 256 |

Table 5: Training hyper-parameters of baseline GPT2 models as well as K-DIAL and RLFC based models in different sizes on WoW and CMU_DoG datasets respectively. **Epoch**, **Batch**, **L.R.**, **Warm.**, **Seq. Len.** denote training epochs, batch size, learning rate, warm-up steps, and maximum sequence length respectively.

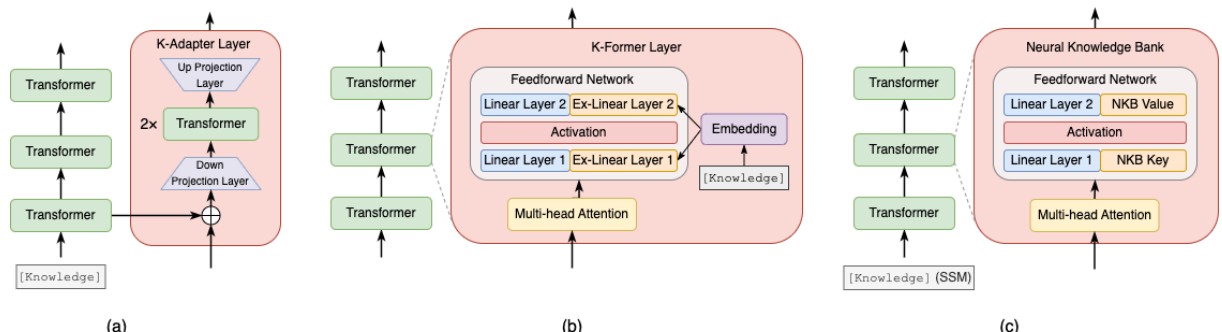

Figure 5: Illustrations of (a) K-ADAPTER; (b) Kformer; (c) Neural Knowledge Bank (NKB) architectures.

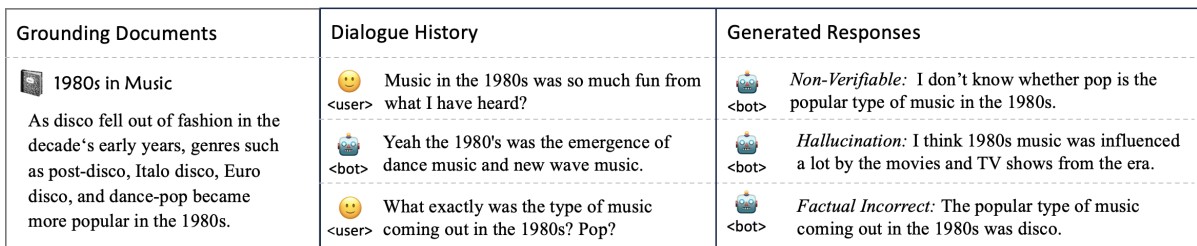

Figure 6

hidden states within the FFN module.

The training procedure for all the knowledge-enhanced models is divided into two stages. First, When injecting knowledge, we froze all the parameters of GPT-2 models and injected factual knowledge (passages of WoW or CMU_DoG datasets) by only updating the parameters in the knowledge module (e.g. K-Adapter, Kformer, and NKB). Then, we fine-tune the knowledge-enhanced models for the two knowledge-grounded dialogue tasks respectively.

| Model | Subset | KF1 | BERT. | Verif. | Hallu. | Fact. | BLEU | F1 | RL. |
|---|---|---|---|---|---|---|---|---|---|
| GPT2-M | Seen | 47.25 | 38.44 | 13.70 | 11.01 | 5.81 | 28.11 | 60.99 | 8.16 |
| + K-DIAL | Seen | 50.07 | 42.23 | 16.77 | 11.62 | 9.39 | 28.26 | 61.63 | 8.97 |
| + RLFC | Seen | 48.02 | 40.56 | 18.91 | 11.25 | 8.57 | 25.82 | 59.49 | 6.37 |
| GPT2-M | Unseen | 45.78 | 38.06 | 13.64 | 10.87 | 5.67 | 26.07 | 60.67 | 7.50 |
| + K-DIAL | Unseen | 48.66 | 41.71 | 16.57 | 11.47 | 9.33 | 28.20 | 61.33 | 8.41 |
| + RLFC | Unseen | 46.45 | 40.12 | 18.77 | 11.06 | 8.31 | 25.04 | 59.03 | 6.11 |

Table 6: Experiments of standard GPT2-M models before and after using K-DIAL and RLFC methods on respective seen and unseen WoW test subsets.

# E Experiments on WoW Seen and Unseen Sets

In this work, whether a topic in the test set has been seen or unseen during training is irrelevant to the effect of knowledge enhancement and alignment. There are no distinct performance improvement differences between seen and unseen test sets of WoW using K-DIAL and RLFC in our early verification experiments as Table 6, which doesn't affect the conclusions. Therefore, we merge the two subsets for all models.