# OpenReview forum: "Improving Factual Consistency for Knowledge-Grounded Dialogue Systems via Knowledge Enhancement and Alignment"
_EMNLP/2023/Conference — EMNLP 2023 Findings_

### Official Review · Reviewer_RZpp · 2023-08-01

**Soundness:** 3

**Excitement:**

2: Mediocre: This paper makes marginal contributions (vs non-contemporaneous work), so I would rather not see it in the conference.

**Paper Topic And Main Contributions:**

This work studies the knowledge-grounded dialogue response generation and focuses on improving factual consistency.

In knowledge-grounded dialogue response generation,  the response generation is guided by the context and the retrieved knowledge. The authors have assumed that such two inputs are always correct and then attributed the reason to the backbone PLMs because the prior knowledge in PLMs learned from large-scale unlabeled corpus might be incomplete.

Consequently, this work first explicitly adds a new FFN unit to the Transformer to boost the prediction of factual knowledge and then proposes a reinforcement learning process to improve the factual consistency with a binary classifier.

Update：I have updated my soundness score to 3 due to the additional experiments.

**Questions For The Authors:**

see my Reasons To Reject

**Reasons To Accept:**

1. This work has identified an urgent issue in the research of knowledge-grounded dialogue generation.

2. The evaluation design of this work is comprehensive and strong because massive metrics are introduced and adopted in this work. This evaluation design could provide a demonstration for relevant researchers.

**Reasons To Reject:**

1. The first KDial can not improve factual consistency.  It's just a technique that makes the model more active toward generating entity words. To cover up this issue, the authors made a strong but not practical statement that the given knowledge is always correct. In fact, it is hard to retrieve the totally correct knowledge. Meanwhile, even if we do not consider this statement, I think the copy mechanism will be more powerful in leveraging the given knowledge compared to using extended FNNs.  Using extended FNNs can be regarded as a simplified copy net.

2. The second RLFC can improve factual consistency, but the idea is a little incremental.

3. This work is a little hard to follow, and thus this work should improve the organization. For example, 1) the description heavily relies on the given figure, but the figure is ambiguous; 2) the detail of K-DIAL is not clear.

4. Although both KDial and +RLFC have shown performance improvements, this work did not compare to any other related solutions besides the naive GPT-2.

**Reproducibility:**

4: Could mostly reproduce the results, but there may be some variation because of sample variance or minor variations in their interpretation of the protocol or method.

**Reviewer Confidence:**

4: Quite sure. I tried to check the important points carefully. It's unlikely, though conceivable, that I missed something that should affect my ratings.

---

> ### Author Rebuttal · Authors · 2023-08-29
>
> We thank the reviewer for the valuable questions. We apologize for the ambiguities in the submitted manuscript. The comments are clarified as follows.
>
> 1. The first K-Dial can not improve factual consistency. It's just a technique that makes the model more active toward generating entity words. To cover up this issue, the authors made a strong but not practical statement that the given knowledge is always correct. In fact, it is hard to retrieve the totally correct knowledge. Meanwhile, even if we do not consider this statement, I think the copy mechanism will be more powerful in leveraging the given knowledge compared to using extended FNNs. Using extended FNNs can be regarded as a simplified copy net.
>
> We thank the reviewer for the insightful comment!
> The reviewer has raised specific questions regarding the efficacy of the proposed K-Dial to improve factual consistency for knowledge-grounded dialogue systems (KDS). We would like to clarify that
>
> - First, the factual consistency in KDS indicates "accurately portrays the input knowledge (assuming the provided knowledge is correct)" as defined in prior research [[1]](https://arxiv.org/abs/2110.05456) which makes a distinction between "factual consistency" and "factual correctness".
>     Factual consistency denotes whether the response utilizes and follows the provided knowledge, where the similar definitions of "attributed" and "supported" are used in [[2]](https://aclanthology.org/2022.tacl-1.62/) and [[3]](https://aclanthology.org/2022.acl-long.263) respectively.
>     For simplicity, we directly leverage the gold knowledge of WoW and CMU\_DoG thus the input knowledge is naturally correct in this work.
>     Therefore, activating the entity words expressions of given knowledge in responses is an effective way to improve factual consistency.
> - Second, The proposed K-Dial is not just a copy net.
>     The K-Dial can better learn and understand on semantics of the knowledge snippet and dialogue contexts by taking advantage of FFN's ability to activate the factual knowledge outputs in provided evidence that are highly related to the dialogue context.
>     The copy mechanism cannot learn to understand the semantic relations between the provided knowledge and dialogue contexts.
>
> 2. The second RLFC can improve factual consistency, but the idea is a little incremental.
>
> We thank the reviewer for the comment.
> We would like to clarify that the proposed RLFC for KDS is specialized in two aspects:
>
> - We collect several released high-quality human-annotated datasets for reward model training.
>     The datasets collect a large number of instances of evaluating the factual consistency of KDS in systematically fine-grained metrics including Verification, Hallucination, and Factual Correctness [[1]](https://arxiv.org/abs/2110.05456).
>     The reward model benefits from such high-quality data to better align with the factual consistency preference of human beings.
>     Therefore, RLFC demonstrates significant performance improvements in factual consistency on KDS tasks.
> - The RLFC implicitly adjusts the FFN's expression, which is a complementarity of K-Dial which explicitly enhances the factual knowledge expressions of FFNs.
>     The supreme performance improvements over baseline GPT2 models are obtained using the combination of RLFC and K-Dial methods.
>
> 3. This work is a little hard to follow, and thus this work should improve the organization. For example, 1) the description heavily relies on the given figure, but the figure is ambiguous; 2) the detail of K-Dial is not clear.
>
> We thank the reviewer for the comment.
> As the training procedure of the proposed method contains several stages that are a bit complex, all the information is difficult to present in one compressed figure.
> We would like to try to revise the figure details and make it clear in the following version.
> We will also supplement the additional implementation details of K-Dial in lines 237-244 in the submitted manuscript for assistance as the following quoted texts.
>
> "The training process of the K-Dial framework is specified in two steps. First, all the parameters of the original PLM are frozen, and the loss $\mathcal L_{K-Dial}$ is only calculated over the knowledge entities in $\boldsymbol Y$ to optimize the parameters of the extended FFNs.
> Afterward, we further adapt the knowledge-enhanced model on the knowledge-grounded dialogue datasets using supervised fine-tuning of Equation (1) while keeping the parameters in extended FFNs fixed.
> The word embedding dimension and hidden size of the extended FFN module are set equal to the corresponding Transformer FFN layers.
> Noted that the K-Dial framework is only applied on the top 3 layers of the model in our experiments."
>
> We present the related hyper-parameters of the baseline model, two stages of K-Dial to train extended FFNs and the full model, and RLFC on both WoW and CMU\_DoG datasets as below.
>
> | Model                           | Dataset  | Epoch | Batch Size | Learning Rate | Warm-up Steps | Sequence Length |
> |-------------------------------------|--------------|-----------|----------------|-------------------|-------------------|---------------------|
> | GPT2-M                              | WoW      | 4         | 16             | 6e-5              | 2k                | 256                 |
> | GPT2-M+ K-Dial (FFNs)        | WoW      | 2         | 32             | 6e-5              | 2k                | 256                 |
> | GPT2-M+ K-Dial (Full GPT2)  | WoW      | 3         | 16             | 6e-5              | 2k                | 256                 |
> | GPT2-M+RLFC                         | WoW      | 3         | 16             | 1e-5              | -                 | 256                 |
> | GPT2-L                              | WoW      | 4         | 8              | 6e-5              | 4k                | 256                 |
> | GPT2-L+K-Dial (FFNs)       | WoW      | 2         | 24             | 6e-5              | 4k                | 256                 |
> | GPT2-L+K-Dial (Full GPT2)  | WoW      | 3         | 8              | 6e-5              | 4k                | 256                 |
> | GPT2-L+RLFC                         | WoW      | 3         | 8              | 1e-5              | -                 | 256                 |
> | GPT2-XL                             | WoW      | 4         | 4              | 6e-5              | 8k                | 256                 |
> | GPT2-XL+K-Dial (FFNs)      | WoW      | 2         | 16             | 6e-5              | 8k                | 256                 |
> | GPT2-XL+K-Dial (Full GPT2) | WoW      | 3         | 4              | 6e-5              | 8k                | 256                 |
> | GPT2-XL+RLFC                        | WoW      | 3         | 4              | 1e-5              | -                 | 256                 |
> | GPT2-M                              | CMU\_DoG | 4         | 8              | 1e-5              | 4k                | 512                 |
> | GPT2-M+K-Dial (FFNs)       | CMU\_DoG | 2         | 32             | 1e-5              | 4k                | 512                 |
> | GPT2-M+K-Dial (Full GPT2)  | CMU\_DoG | 3         | 8              | 1e-5              | 4k                | 512                 |
> | GPT2-L                              | CMU\_DoG | 4         | 4              | 1e-5              | 8k                | 512                 |
> | GPT2-L+K-Dial (FFNs)       | CMU\_DoG | 2         | 16             | 1e-5              | 8k                | 512                 |
> | GPT2-L+K-Dial (Full GPT2)  | CMU\_DoG | 3         | 4              | 1e-5              | 8k                | 512                 |
> | GPT2-XL                             | CMU\_DoG | 4         | 4              | 1e-5              | 8k                | 512                 |
> | GPT2-XL+K-Dial (FFNs)      | CMU\_DoG | 2         | 16             | 1e-5              | 8k                | 512                 |
> | GPT2-XL+K-Dial (Full GPT2) | CMU\_DoG | 3         | 4              | 1e-5              | 8k                | 512                 |
>
>
> 4. Although both K-Dial and +RLFC have shown performance improvements, this work did not compare to any other related solutions besides the naive GPT-2.
>
> We thank the reviewer for the comment.
> For other related solutions, to the best of our knowledge, this work is the first to propose to improve factual consistency for knowledge-grounded dialogue systems (KDS).
> Previous related work that investigates the factual consistency of KDS focuses on evaluation method [[7]](https://aclanthology.org/2021.emnlp-main.619) or datasets [[1]](https://arxiv.org/abs/2110.05456), [[2]](https://aclanthology.org/2022.tacl-1.62/), [[3]](https://aclanthology.org/2022.acl-long.263).
> Therefore, there is no direct research to compare with our proposed improvement method except baseline supervised fine-tuned GPT-2 models.
>
> Nevertheless, for the knowledge-enhancing method K-Dial, we still conduct experiments on several knowledge injection and enhancement methods for PLMs including K-Adapter [[4]](https://doi.org/10.18653/v1/2021.findings-acl.121), Kformer [[5]](https://arxiv.org/abs/2201.05742), and Neural Knowledge Bank [[6]](https://arxiv.org/abs/2208.00399) and adapt on KDS tasks as presented in Table 4 in the submitted manuscript.
> Experimental results suggest that our proposed K-Dial outperforms other baseline methods with improvements in factual consistency with grounding knowledge without sacrificing the dialogue qualities.
>
> **Reference**
> - [1] [Rome was built in 1776: A case study on factual correctness in knowledge-grounded response generation](https://arxiv.org/abs/2110.05456)
> - [2] [Evaluating Attribution in Dialogue Systems: The BEGIN Benchmark](https://aclanthology.org/2022.tacl-1.62/)
> - [3] [DialFact: A benchmark for fact-checking in dialogue](https://aclanthology.org/2022.acl-long.263)
> - [4] [K-adapter: Infusing knowledge into pre-trained models with adapters](https://doi.org/10.18653/v1/2021.findings-acl.121)
> - [5] [Kformer: Knowledge injection in transformer feed-forward layers](https://arxiv.org/abs/2201.05742)
> - [6] [Neural knowledge bank for pretrained transformers](https://arxiv.org/abs/2208.00399)
> - [7] [q 2: Evaluating factual consistency in knowledge-grounded dialogues via question generation and question answering](https://aclanthology.org/2021.emnlp-main.619)

---

### Official Review · Reviewer_j1xH · 2023-08-04

**Soundness:** 4

**Excitement:**

4: Strong: This paper deepens the understanding of some phenomenon or lowers the barriers to an existing research direction.

**Paper Topic And Main Contributions:**

The paper proposed K-DIAL (knowledge-enhanced dialogue generation) and RLFC (reinforcement learning for factual consistency) to enhance the factual knowledge expression capability of FFNs in Transformers by introducing extended FFNs and aligning the responses with gold knowledge respectively.

**Reasons To Accept:**

1. By introducing extended FFNs which can be viewed as key-value memories, K-DIAL can effectively update the knowledge into the network parameters.

2. Inspired by RLHF, RLFC effectively encourages the model to output knowledge-consistent responses.

3. Experimental results show the effectiveness of the proposed method.

**Reasons To Reject:**

1. As mentioned in Section 4.1.1, the enhancement achieved by the (GPT2-XL) method is relatively limited when applied to larger models. As stated in Section Limitation, methods have not yet been validated on larger scale language models.


**Reproducibility:**

4: Could mostly reproduce the results, but there may be some variation because of sample variance or minor variations in their interpretation of the protocol or method.

**Reviewer Confidence:**

4: Quite sure. I tried to check the important points carefully. It's unlikely, though conceivable, that I missed something that should affect my ratings.

---

> ### Author Rebuttal · Authors · 2023-08-28
>
> We thank the reviewer for the insightful comment. Due to the constraints of computation resources, experiments were limited to be conducted on Large-scale LMs which presently entail expensive costs to fully fine-tune the parameters.
>
> Regarding the significant improvements demonstrated by our proposed methodologies which prove the efficiency, we are actively committed to an ensuing endeavor aimed at the extension of K-Dial and RLFC on widely used LLMs like LLaMA and Alpaca in future work.

---

### Official Review · Reviewer_BZCd · 2023-08-10

**Soundness:** 2

**Excitement:**

2: Mediocre: This paper makes marginal contributions (vs non-contemporaneous work), so I would rather not see it in the conference.

**Paper Topic And Main Contributions:**

This paper focuses on improving the factual consistency of PLM-based dialogue systems given the grounding factual knowledge. The authors attribute the factual inconsistency to the innate limitations of PLMs and propose two solutions to enhance the factual consistency of PLMs: 1) They first explicitly extend FFN modules in Transformers to express factual knowledge. 2) They use Reinforcement Learning to implicitly align generated responses with the grounding knowledge for factual consistency. And the authors conduct experiments on WoW and CMU_DoG two datasets to demonstrate the effectiveness of their proposed methods in improving the factual consistency of PLMs.

**Reasons To Accept:**

1. The authors attribute the factual inconsistency to the innate limitations of PLMs and propose two solutions to enhance the factual consistency of PLMs, which is reasonable.
2. The writing is clear and easy to follow.

**Reasons To Reject:**

1.	The motivation of this paper is not exciting, and the paper does not provide valuable insights. The authors claim that they focus on promoting the FFN modules in Transformers to improve factual consistency. To my best knowledge, there are lots of related works, such as knowledge editing (Refs [1]~[4]). Why is there no explicit comparison of their method (i.e. K-Dial) with existing related works? I think the author should discuss the difference between the proposed method and other relevant works in the Related work and experiment, which can provide valuable insights into the subfield of factual consistency.
2.	Some important information is missing.
a)	The training details of the proposed RLFC. In Section 2.4, although the authors show the overall workflow of RLFC, they do not provide information about reward design and optimization objectives. This omission makes it difficult for researchers to reproduce their work.
b)	The rating details about human evaluations in the experiments section. In line 479, the authors do not tell the rating details about human evaluations, such as score range and criteria. This makes it difficult for readers to assess the credibility of human evaluations.
3.	The experiments are not comprehensive and there is negligence.
a)	Experiments do not appear to be sufficient to convince of the superiority of the proposed method. There is no reasonable baselines for comparison. Why do not the authors compare their proposed method with any relevant baselines?
b)	The experiments on CMU_DoG dataset are not comprehensive, lacking RLFC, RLFC+K-DIAL, ChatGPT evaluation and human evaluation on GPT-2 models. Moreover, the authors do not conduct the comparisons and Ablation study shown in Section 4.2.
c)	The experimental results are not consistency. In Table 4, the GPT2-M + K-DIAL achieves 49.36 score in KF1 metric. However, in Table 1, the same model (i.e., GPT2-M+K-DIAL) achieves 48.36 score in KF1 metric.


Refs:
[1]: https://openreview.net/pdf?id=Plr5l7r0jY6
[2]: https://arxiv.org/pdf/2012.00363.pdf
[3]: https://arxiv.org/pdf/2205.11482v3.pdf
[4]: https://openreview.net/pdf?id=ngCT1EelZk


**Reproducibility:**

3: Could reproduce the results with some difficulty. The settings of parameters are underspecified or subjectively determined; the training/evaluation data are not widely available.

**Reviewer Confidence:**

3: Pretty sure, but there's a chance I missed something. Although I have a good feel for this area in general, I did not carefully check the paper's details, e.g., the math, experimental design, or novelty.

---

> ### Author Rebuttal · Authors · 2023-08-29
>
> We thank the reviewer for the valuable questions. We apologize for the ambiguities in the submitted manuscript. The comments are clarified as follows.
>
>
>
> 1. The motivation of this paper is not exciting, and the paper does not provide valuable insights. The authors claim that they focus on promoting the FFN modules in Transformers to improve factual consistency. To my best knowledge, there are lots of related works, such as knowledge editing. Why is there no explicit comparison of their method (i.e. K-Dial) with existing related works? I think the author should discuss the difference between the proposed method and other relevant works in the Related work and experiment, which can provide valuable insights into the subfield of factual consistency.
>
> We thank the reviewer for the insightful comment and for providing valuable related works.
> The reviewer has raised concerns about the lack of baseline comparisons with some knowledge editing methods for our proposed K-Dial.
> We would like to clarify that
>
> - First, we have presented several baseline comparisons of several knowledge injection or enhancement methods including K-Adapter [[1]](https://doi.org/10.18653/v1/2021.findings-acl.121), Kformer [[2]](https://arxiv.org/abs/2201.05742), and Neural Knowledge Bank [[3]](https://arxiv.org/abs/2208.00399) which are adapted on KDS tasks in Table 4 in the submitted paper.
>     Experimental results demonstrate our proposed K-Dial outperforms other baseline methods on both factual consistency and dialogue qualities, suggesting that K-Dial can take advantage of FFN's ability to express factual knowledge given specific input patterns (or prompts) of the concatenated external knowledge and dialogue contexts.
>
> - Second, the reviewer's provided works are knowledge-editing methods that seek and edit alternative parameters to calibrate the incorrect knowledge and make new predictions on revised instances such as [[4]](https://aclanthology.org/2022.findings-emnlp.438/).
>     For KDS, we focus more on integrating the external knowledge base to generate responses in a knowledgeable way by dialogue models.
>     Therefore, the knowledge enhancement method which incorporates external knowledge to PLMs, and mostly into additional modules like adapters [[1]](https://doi.org/10.18653/v1/2021.findings-acl.121) is more appropriate for KDS tasks.
>
>
>
>
> 2. Some important information is missing. a) The training details of the proposed RLFC. In Section 2.4, although the authors show the overall workflow of RLFC, they do not provide information about reward design and optimization objectives. This omission makes it difficult for researchers to reproduce their work. b) The rating details about human evaluations in the experiments section. In line 479, the authors do not tell the rating details about human evaluations, such as score range and criteria. This makes it difficult for readers to assess the credibility of human evaluations.
>
> We thank the reviewer for the comment.
> We would like to clarify that
>
> - The reward design and optimization objectives are all provided in Section 2.4 in the submitted manuscript.
>     The reward model $R(\cdot)$ is a BERT-based factual consistency NLI classifier described in line 273-288 and line 339-342.
>     The reward score $r_1$ is calculated by $r_1 = R(\boldsymbol K, \boldsymbol Y^{(1)})$ where $\boldsymbol K$ and $\boldsymbol Y^{(1)}$ denote the grounding knowledge and the response generated by the policy model.
> The optimization objective $r$ is determined by $r=r_1+ r_2$.
>     $r_1$ is the reward score and $r_2$ is the KL divergence between policy model and reference model generated responses $\boldsymbol Y^{(1)}$ and $\boldsymbol Y^{(2)}$, which is introduced in line 289-308 and Figure 3.
>     The hyper-parameters of PPO algorithm are referred to the trl PPOConfig recipe [[5]](https://github.com/huggingface/trl/blob/main/trl/trainer/ppo\_config.py).
>
> - The Human Evaluations are asked to rate each quality on a two-level Likert scale from 0 (Not Flunet, Not Coherent, Inconsistent) to 1 (Flunet, Coherent, Consistent) to evaluate the Fluency, Coherence, and Factual Consistency of generated responses as described in line 479-491. The averaged results by the human evaluations are reported in Table 2 in paper.
>
>
> 3. The experiments are not comprehensive and there is negligence. a) Experiments do not appear to be sufficient to convince of the superiority of the proposed method. There are no reasonable baselines for comparison. Why do not the authors compare their proposed method with any relevant baselines? b) The experiments on the CMU\_DoG dataset are not comprehensive, lacking RLFC, RLFC+K-Dial, ChatGPT evaluation, and human evaluation on GPT-2 models. Moreover, the authors do not conduct the comparisons and Ablation study shown in Section 4.2. c) The experimental results are not consistent. In Table 4, the GPT2-M + K-Dial achieves 49.36 score in KF1 metric. However, in Table 1, the same model (i.e., GPT2-M+K-Dial) achieves 48.36 score in KF1 metric.
>
>
> We thank the reviewer for the comment.
> We would like to clarify that
>
> - For baseline comparisons, to the best of our knowledge, this work is the first to propose to improve factual consistency for KDS.
>     Previous related works that investigate the factual consistency of KDS only focus on evaluation method [[6]](https://aclanthology.org/2021.emnlp-main.619) or datasets [[7]](https://arxiv.org/abs/2110.05456), [[8]](https://aclanthology.org/2022.tacl-1.62/), [[9]](https://aclanthology.org/2022.acl-long.263).
>     Therefore, there is no direct improvement method to be compared.
>
>     Nevertheless, for the knowledge-enhancing method K-Dial, we still conduct experiments on several knowledge injection and enhancement methods for NLG tasks, including K-Adapter [[1]](https://doi.org/10.18653/v1/2021.findings-acl.121), Kformer [[2]](https://arxiv.org/abs/2201.05742), and Neural Knowledge Bank [[3]](https://arxiv.org/abs/2208.00399), which are adapted on KDS tasks as baselines presented in Table 4.
>     The results suggest that our proposed K-Dial outperforms other baseline methods with improvements in factual consistency while without sacrificing the dialogue qualities.
>  -  For RLFC related experiments, the reward model is trained on NLI datasets, but we lack of high-quality human-annotated factual consistency NLI data on CMU\_DoG.
>     The high-quality fine-grained NLI datasets to evaluate factual consistency including Verification, Hallucination, and Factual Correctness are only collected and annotated on WoW dataset [[7]](https://arxiv.org/abs/2110.05456), [[8]](https://aclanthology.org/2022.tacl-1.62/), [[9]](https://aclanthology.org/2022.acl-long.263).
>     Directly transferring the NLI model on WoW to CMU\_DoG for RLFC training does not obtain comparable performance improvements.
>     Presumably these fine-grained factual consistency metrics to evaluate KDS are highly dependent on specific datasets.
>     We are trying to explore a more general NLI model for factual consistency evaluation that can adapt to different datasets in the revised manuscript.
>
> - For ChatGPT and Human Evaluations, as well as the comparisons and ablation studies on CMU\_DoG, due to the page limitation, they cannot be presented in the submitted version. We will supplement all the experiments in future work.
>
> - We apologize for the typo. The correct result is 48.36 of GPT2-M+K-Dial in Table 1 in KF1 metric.
>
>
> **Reference**
> - [1] [K-adapter: Infusing knowledge into pre-trained models with adapters](https://doi.org/10.18653/v1/2021.findings-acl.121)
> - [2] [Kformer: Knowledge injection in transformer feed-forward layers](https://arxiv.org/abs/2201.05742)
> - [3] [Neural knowledge bank for pretrained transformers](https://arxiv.org/abs/2208.00399)
> - [4] [Calibrating factual knowledge in pretrained language models](https://aclanthology.org/2022.findings-emnlp.438/)
> - [5] [trl](https://github.com/huggingface/trl/blob/main/trl/trainer/ppo\_config.py)
> - [6] [q 2: Evaluating factual consistency in knowledge-grounded dialogues via question generation and question answering](https://aclanthology.org/2021.emnlp-main.619)
> - [7] [Rome was built in 1776: A case study on factual correctness in knowledge-grounded response generation](https://arxiv.org/abs/2110.05456)
> - [8] [Evaluating Attribution in Dialogue Systems: The BEGIN Benchmark](https://aclanthology.org/2022.tacl-1.62/)
> - [9] [DialFact: A benchmark for fact-checking in dialogue](https://aclanthology.org/2022.acl-long.263)

---

### Official Review · Reviewer_waXu · 2023-08-12

**Soundness:** 3

**Excitement:**

2: Mediocre: This paper makes marginal contributions (vs non-contemporaneous work), so I would rather not see it in the conference.

**Paper Topic And Main Contributions:**

This paper proposes extended FFN modules in Transformer and reinforcement learning for knowledge-based dialogue generation. The FFN modules aim to ground knowledge-related token to response. The reinforcement learning is inspired by RLHF, reinforcing the model to prefer factually consistent responses using NLI metrics. To validate the effectiveness of the proposed approach, the study conducts experiments on the WoW and CMU_DoG datasets.

**Questions For The Authors:**

* Could you provide the number of parameters used for each approach in Table 4?
* Is the result for CMU_DoG still consistent, similar to that of Table 1 (RLFC+K-DIAL)? Could you also report that result?
* Additional clarification is needed regarding the model's training process (Lines 241-244). The current information lacks specific training details, such as the number of epochs/steps dedicated to training FFNs and the full model. Moreover, it is unclear whether lambda is utilized for KL-penalty. Furthermore, during the training of RLFC, could you specify whether the training only focuses on the FFNs or LM?
* Was there a performance difference between the seen and unseen of WoW?
* What specific scale was employed for the Human evaluation? (For instance, a 3-point Likert scale)
* How are the final models for each row determined in Table 1? Did you use early stopping or fixed training epoch?

**Reasons To Accept:**

The experimental results demonstrate the effectiveness of the combination of FFNs and RL in the generation of knowledge-based dialogues.

**Reasons To Reject:**

There are performance gains associated with the proposed components; however, I am concerned about whether these gains arise from an increase in training steps or the use of additional parameters. K-DIAL and K-DIAL($L_{CE}$) were comparable in most metrics, indicating that Equation 3 does not appear to be advantageous. By addressing the following questions, you might be able to resolve my concerns.

**Reproducibility:**

2: Would be hard pressed to reproduce the results. The contribution depends on data that are simply not available outside the author's institution or consortium; not enough details are provided.

**Reviewer Confidence:**

3: Pretty sure, but there's a chance I missed something. Although I have a good feel for this area in general, I did not carefully check the paper's details, e.g., the math, experimental design, or novelty.

---

> ### Author Rebuttal · Authors · 2023-08-29
>
> We thank the reviewer for the valuable questions.
> We apologize for the ambiguities in the submitted manuscript.
> The questions are addressed as follows.
>
> **Questions For The Authors:**
>
> 1. Could you provide the number of parameters used for each approach in Table 4?
>
> We apologize for the missing additional parameter information in the submitted paper.
> The updated Table 4 with the number of parameters is as follows
>
> | Model                       | # Param | KF1 | BERTScore | Verif. | Hallu. | Fact. | ChatGPT | $Q^2$ | BLEU | F1 | RL. |
> |--------------------------------------------------|----------------|-----------|-----------------|--------------|--------------|-------------|---------------|--------------------|------------|----------|-----------|
> | GPT2-M                                           | 355M           | 46.51     | 38.25           | 13.67        | 10.94        | 5.74        | 26.25         | 55.55              | 27.09      | 60.83    | 7.83      |
> | + K-Adapter                            | 377M           | 46.96     | 39.32           | 14.66        | 10.35        | 7.64        | 28.75         | 60.36              | 26.03      | 59.17    | 7.06      |
> | + K-Former                                     | 361M           | 47.43     | 39.91           | 15.51        | 11.50        | 8.37        | 26.25         | 57.87              | 26.28      | 59.08    | 8.14      |
> | + NKB                                          | 361M           | 46.60     | 38.74           | 13.49        | 9.51         | 6.35        | 25.00         | 58.88              | 25.30      | 60.55    | 6.84      |
> | + K-Dial                              | 361M           | 48.36     | 41.97           | 16.67        | 11.54        | 9.36        | 32.50         | 64.47              | 28.33      | 61.48    | 8.69      |
> | + K-Dial ($\mathcal L_{\mathrm{CE}}$) | 361M           | 49.03     | 41.66           | 16.50        | 11.23        | 9.48        | 32.50         | 62.34              | 29.36      | 61.06    | 8.40      |
>
> where the **# Param** denotes the number of total model parameters.
> Note that the details of model designs of the knowledge-enhanced methods are presented in Appendix D in the paper.
>
>
> 2. Is the result for CMU\_DoG still consistent, similar to that of Table 1 (RLFC+ K-Dial)? Could you also report that result?
>
> For RLFC-related experiments, the reward model is trained on NLI datasets, but we lack high-quality human-annotated factual consistency NLI data on CMU\_DoG.
> The high-quality fine-grained NLI datasets to evaluate factual consistency including Verification, Hallucination, and Factual Correctness, are only collected and annotated on WoW dataset [[1]](https://arxiv.org/abs/2110.05456), [[2]](https://aclanthology.org/2022.tacl-1.62/), [[3]](https://aclanthology.org/2022.tacl-1.62/).
>
> Directly transferring the NLI model on WoW to CMU\_DoG for RLFC training cannot obtain comparable performance improvements.
> Presumably these fine-grained factual consistency metrics to evaluate KDS are highly dependent on specific datasets.
> We are trying to explore a more general NLI model for factual consistency evaluation that can be adapted to different datasets in future work.
>
> 3. Additional clarification is needed regarding the model's training process (Lines 241-244). The current information lacks specific training details, such as the number of epochs/steps dedicated to training FFNs and the full model. Moreover, it is unclear whether lambda is utilized for KL-penalty. Furthermore, during the training of RLFC, could you specify whether the training only focuses on the FFNs or LM?
>
> - For additional information on the training process of K-Dial (Lines 241-244), the related hyper-parameters of two stages to train extended FFNs and the full parameters of GPT2 models on both WoW and CMU\_DoG datasets are listed below.
>     All the models are trained to converge in the fixed training epochs.
>
> | Model                           | Dataset  | Epoch | Batch Size | Learning Rate | Warm-up Steps | Sequence Length |
> |-------------------------------------|--------------|-----------|----------------|-------------------|-------------------|---------------------|
> | GPT2-M                              | WoW      | 4         | 16             | 6e-5              | 2k                | 256                 |
> | GPT2-M+ K-Dial (FFNs)        | WoW      | 2         | 32             | 6e-5              | 2k                | 256                 |
> | GPT2-M+ K-Dial (Full GPT2)  | WoW      | 3         | 16             | 6e-5              | 2k                | 256                 |
> | GPT2-M+RLFC                         | WoW      | 3         | 16             | 1e-5              | -                 | 256                 |
> | GPT2-L                              | WoW      | 4         | 8              | 6e-5              | 4k                | 256                 |
> | GPT2-L+K-Dial (FFNs)       | WoW      | 2         | 24             | 6e-5              | 4k                | 256                 |
> | GPT2-L+K-Dial (Full GPT2)  | WoW      | 3         | 8              | 6e-5              | 4k                | 256                 |
> | GPT2-L+RLFC                         | WoW      | 3         | 8              | 1e-5              | -                 | 256                 |
> | GPT2-XL                             | WoW      | 4         | 4              | 6e-5              | 8k                | 256                 |
> | GPT2-XL+K-Dial (FFNs)      | WoW      | 2         | 16             | 6e-5              | 8k                | 256                 |
> | GPT2-XL+K-Dial (Full GPT2) | WoW      | 3         | 4              | 6e-5              | 8k                | 256                 |
> | GPT2-XL+RLFC                        | WoW      | 3         | 4              | 1e-5              | -                 | 256                 |
> | GPT2-M                              | CMU\_DoG | 4         | 8              | 1e-5              | 4k                | 512                 |
> | GPT2-M+K-Dial (FFNs)       | CMU\_DoG | 2         | 32             | 1e-5              | 4k                | 512                 |
> | GPT2-M+K-Dial (Full GPT2)  | CMU\_DoG | 3         | 8              | 1e-5              | 4k                | 512                 |
> | GPT2-L                              | CMU\_DoG | 4         | 4              | 1e-5              | 8k                | 512                 |
> | GPT2-L+K-Dial (FFNs)       | CMU\_DoG | 2         | 16             | 1e-5              | 8k                | 512                 |
> | GPT2-L+K-Dial (Full GPT2)  | CMU\_DoG | 3         | 4              | 1e-5              | 8k                | 512                 |
> | GPT2-XL                             | CMU\_DoG | 4         | 4              | 1e-5              | 8k                | 512                 |
> | GPT2-XL+K-Dial (FFNs)      | CMU\_DoG | 2         | 16             | 1e-5              | 8k                | 512                 |
> | GPT2-XL+K-Dial (Full GPT2) | CMU\_DoG | 3         | 4              | 1e-5              | 8k                | 512                 |
>
> - For the training details of RLFC, lambda is set to 0.95 for KL-penalty.
>     All the hyper-parameters related to PPO are provided default values by the trl PPOConfig recipe [[4]](https://github.com/huggingface/trl/blob/main/trl/trainer/ppo\_config.py) except the epoch, learning rate, and batch size which are provided as above.
> - During the training of RLFC, all the parameters of GPT2 are updated.
>     We also experimented by applying RLFC on FFNs only but got poor performance results.
>     Since the FFNs are mainly responsible for the factual knowledge expressions [[5]](https://aclanthology.org/2021.emnlp-main.446), only updating parameters in FFN cannot improve the knowledge-grounded dialogue ability compared with fully fine-tuned baseline models.
>     Therefore, RLFC must be used on full model parameters which can implicitly enhance the FFNs' ability to express the factual knowledge in responses.
>
> 4. Was there a performance difference between the seen and unseen of WoW?
> What specific scale was employed for the Human evaluation? (For instance, a 3-point Likert scale)
>
> - In this work, whether a topic in the test set has been seen or unseen during training is irrelevant to the effect of knowledge enhancement and alignment.
>     There are no distinct performance improvement differences between seen and unseen test sets of WoW using K-Dial and RLFC in our early verification experiments as below, which doesn't affect the conclusions.
>     Therefore, we merge the results of two subsets for all models.
>
> | Model | Subset | KF1 | BERTScore | Verif. | Hallu. | Fact. | BLEU | F1 | RL. |
> |----------------------------|--------------|-----------|-----------------|--------------|--------------|-------------|------------|----------|-----------|
> | GPT2-M                     | Seen         | 47.25     | 38.44           | 13.70        | 11.01        | 5.81        | 28.11      | 60.99    | 8.16      |
> | + K-Dial         | Seen         | 50.07     | 42.23           | 16.77        | 11.62        | 9.39        | 28.26      | 61.63    | 8.97      |
> | + RLFC                   | Seen         | 48.02     | 40.56           | 18.91        | 11.25        | 8.57        | 25.82      | 59.49    | 6.37      |
> | GPT2-M                     | Unseen       | 45.78     | 38.06           | 13.64        | 10.87        | 5.67        | 26.07      | 60.67    | 7.50      |
> | + K-Dial        | Unseen       | 48.66     | 41.71           | 16.57        | 11.47        | 9.33        | 28.20      | 61.33    | 8.41      |
> | + RLFC                   | Unseen       | 46.45     | 40.12           | 18.77        | 11.06        | 8.31        | 25.04      | 59.03    | 6.11      |
>
> - The Human Evaluations are asked to rate each quality on a two-level Likert scale from 0 (Not Flunet, Not Coherent, Inconsistent) to 1 (Flunet, Coherent, Consistent) to evaluate the Fluency, Coherence, and Factual Consistency of generated responses as described in line 479-491.
> The averaged results of the human evaluations are reported in Table 2 in the paper.
>
> 5. How are the final models for each row determined in Table 1? Did you use early stopping or fixed training epoch?
>
> We use the fixed training epoch to determine the final models in Table 1.
> The optimal training strategy to obtain the final models is specialized in two stages.
> First, we train the GPT2 models using RLFC.
> Then K-Dial method is applied to the obtained model.
> The fixed training epoch strategy is adopted on both RLFC and K-Dial stages as presented in the Table in Question 3 in this rebuttal.
>
> **Reference**
>
> - [1] [Rome was built in 1776: A case study on factual correctness in knowledge-grounded response generation](https://arxiv.org/abs/2110.05456)
> - [2] [Evaluating Attribution in Dialogue Systems: The BEGIN Benchmark](https://aclanthology.org/2022.tacl-1.62/)
> - [3] [DialFact: A benchmark for fact-checking in dialogue](https://aclanthology.org/2022.tacl-1.62/)
> - [4] [trl](https://github.com/huggingface/trl/blob/main/trl/trainer/ppo\_config.py)
> - [5] [Transformer feed-forward layers are key-value memories](https://aclanthology.org/2021.emnlp-main.446)

---

### Meta-Review · Area_Chair_Qnt1 · 2023-09-19

**Recommendation:** 3

**Metareview:**

This paper presents K-DIAL which introduces extensions to the FFN in Transformer neural architecture, and apply a reinforcement learning for factual consistency method to align the responses with ground-truth knowledge. The presentation clarity in the paper leaves room for further improvement. While the authors did clarify  some of the important details missing from the original paper during rebuttal, these should be incorporated back into the manuscript.

---

### Decision · Program_Chairs · 2023-10-07

**Decision:**

Accept-Findings

**Comment:**

This paper presents K-DIAL which introduces extensions to the FFN in Transformer neural architecture, and apply a reinforcement learning for factual consistency method to align the responses with ground-truth knowledge. The presentation clarity in the paper leaves room for further improvement. While the authors did clarify  some of the important details missing from the original paper during rebuttal, these should be incorporated back into the manuscript.